# TGF-β as a Key Modulator of Astrocyte Reactivity: Disease Relevance and Therapeutic Implications

**DOI:** 10.3390/biomedicines10051206

**Published:** 2022-05-23

**Authors:** Jian Luo

**Affiliations:** Palo Alto Veterans Institute for Research, VAPAHCS, Palo Alto, CA 94304, USA; jluo@pavir.org

**Keywords:** astrocytes, reactive astrogliosis, TGF-β, traumatic brain injury, stroke, aging, Alzheimer’s disease, Parkinson’s disease, amyotrophic lateral sclerosis, multiple sclerosis, epilepsy

## Abstract

Astrocytes are essential for normal brain development and functioning. They respond to brain injury and disease through a process referred to as reactive astrogliosis, where the reactivity is highly heterogenous and context-dependent. Reactive astrocytes are active contributors to brain pathology and can exert beneficial, detrimental, or mixed effects following brain insults. Transforming growth factor-β (TGF-β) has been identified as one of the key factors regulating astrocyte reactivity. The genetic and pharmacological manipulation of the TGF-β signaling pathway in animal models of central nervous system (CNS) injury and disease alters pathological and functional outcomes. This review aims to provide recent understanding regarding astrocyte reactivity and TGF-β signaling in brain injury, aging, and neurodegeneration. Further, it explores how TGF-β signaling modulates astrocyte reactivity and function in the context of CNS disease and injury.

## 1. Introduction

Astrocytes perform a wide variety of complex and essential functions in maintaining central nervous system (CNS) homeostasis and are pivotal responders to a wide spectrum of pathological insults [1]. Astrocytes respond to insults to the brain by a process referred to as “reactive astrogliosis” and undergo a spectrum of changes in gene expression, as well as morphological, biochemical, metabolic, and physiological remodeling, which ultimately result in the gain of new function(s) or loss or upregulation of homeostatic ones (definition by a recently published consensus statement [1]). Astrocyte reactivity is highly heterogeneous and dynamic, and comprises a graded continuum of context-dependent responses that may result in adaptive or maladaptive effects [2]. Maladaptive astrocytes lose homeostatic functions and may also gain detrimental ones, thus exacerbating ongoing pathology and promoting disease progression [1,2,3,4,5,6,7,8], but the cellular and molecular mechanisms of maladaptive reactivity are not well understood. Dissecting underlying mechanisms and learning how to beneficially modulate astrocyte reactivity are therefore of great importance for astrocyte research.

The transforming growth factor-β (TGF-β) superfamily consists of a large group of pleiotropic cytokines with a wide range of essential functions, including cell development, differentiation, proliferation, and survival; tissue homeostasis, remodeling, and repair; morphogenesis and angiogenesis; and inflammation and immune responses [9,10,11]. Members of the TGF-β superfamily regulate multiple aspects of brain function during development and in the adult brain, and aberrant TGF-β signaling contributes to the pathogenesis of neurological disorders [11,12,13,14]. TGF-β has been identified as a key regulator of astrocyte reactivity and glial scar formation [5,15,16,17]. The manipulation of the TGF-β signaling pathway in astrocytes alters pathological and functional outcomes in models of neurological diseases. Recent advances in genomics and multi-omics have yielded novel insights into astrocyte reactivity and diversity, and TGF-β signaling in health and disease. This review will discuss recent insights into how astrocytes respond and function in response to brain injury, aging, and neurodegeneration, focusing on how TGF-β signaling regulates astrogliosis and affects disease outcome.

## 2. TGF-β Signaling in the Brain

The TGF-β superfamily includes TGF-βs, bone morphogenetic proteins (BMP), growth differentiation factors (GDFs), activins, nodal, inhibins, and anti-Mullerian hormone proteins (Müllerian inhibiting substance) [9,10,11]. TGF-β isoforms (TGF-β1, 2, and 3) are synthesized as protein precursors with a 70–80% homology. They are secreted in a biologically inactive (latent) form as part of a latent complex stored in the extracellular matrix. Latent TGF-β must be activated before binding to the receptors [9,10]. The activation is a critical step in the regulation of TGF-β’s activity. Activators of latent TGF-β include cell surface integrins (α_v_β_6_ and α_v_β_8_), proteases (calpain, cathepsin D, and matrix metalloproteinase), thrombospondin-1 (TSP-1), and reactive oxygen species (ROS) [18] (Figure 1). The TGF-β complex serves as an extracellular sensor and control switch responding to specific signals or perturbations by releasing active TGF-β [19].

Canonical TGF-β signaling is mediated by the binding of TGF-β1-3 to TGF-β receptor types 1 and 2 (TGFBR1/2), which are transmembrane serine/threonine kinase receptors (Figure 1). Upon TGF-β binding, TGFBR1 (also called activin receptor-like kinase 5, ALK5) and TGFBR2 form a heteromeric complex [9,10]. In the receptor complex, TGFBR2 subunits phosphorylate and activate TGFBR1, which phosphorylates SMAD2 and SMAD3. Phosphorylated SMAD2/3 then form a complex with SMAD4 and translocate into the nucleus, where they bind to the Smad binding element (SBE) to regulate the expression of TGF-β/Smad-responsive genes [9,10] (Figure 1). The complex signaling pathway with numerous signaling components presents a great deal of flexibility, as well as complexity, in manipulating TGF-β signaling in a disease context. As such, TGF-β signaling can be manipulated through different strategies, from production, activation, and receptor binding to downstream Smads and target genes (Figure 1).

The SBE has been demonstrated to confer TGF-β/Smad-responsive transcription in numerous in vivo and in vitro assays and has thus been used as a promoter for reporter systems specific to the TGF-β/Smad signaling pathway. We generated SBE-luc mice that express luciferase under the control of an SBE promoter [20,21,22,23]. These mice allow us to monitor the temporal, tissue-specific activation of Smad2/3-dependent signaling in living mice [20,21,22,23]. By analyzing the SBE-luc mice, we discovered that the brain has the highest basal TGF-β signaling among all organs, and that, within the brain, the hippocampus has the highest basal TGF-β signaling among all regions [21,23]. Basal TGF-β signaling is localized primarily to pyramidal neurons [21]. The hippocampus, which plays a critical role in learning and memory, is particularly vulnerable to a variety of insults [24]. These results indicate the importance of TGF-β signaling in maintaining the normal function of the hippocampus. The SBE-luc mice have also helped us to identify endogenous and synthetic modulators of the TGF-β/Smad pathway [25,26].

In the CNS, TGF-βs and their receptors are widely expressed among many cell types and they play an essential role in the differentiation, development, and function of glial, neuronal, and endothelial cells [12]. In astrocytes, TGF-β regulates radical glia (neuroglial progenitor) differentiation and astrocyte formation during development [14]. The expression of glial fibrillary acidic protein (GFAP), an intermediate filament protein commonly used as a classical marker for mature astrocytes, is tightly regulated during development and under pathological conditions [27,28]. TGF-β signaling is one of the main pathways that regulate GFAP promoter activity and expression [14]. Neurons, especially during the embryonic period, secrete TGF-β1 that induces GFAP expression in astrocytes [27,28]. Interestingly, by regulating TGF-β1 synthesis and secretion, neurons at different developmental stages and from different regions exert different effects on GFAP expression [27,28]. On the other hand, astrocyte-derived TGF-β mediates crosstalk between astrocytes and other cells in the brain and modulates their functions. The overexpression of TGF-β1 selectively in astrocytes reduces neurogenesis and astrogenesis in the GFAP-TGF-β1 mice [29]. During inflammation, astrocytes alone [30] or cooperating neurons [31] release TGF-β to promote microglial homeostasis and repress inflammatory responses. Therefore, TGF-β1 is not only a key regulator of astrocyte formation and maturation, but also an effector of astrocyte function, mediating crosstalk between astrocytes and other cells in the CNS.

## 3. Astrocytes Reactivity

Astrocytes are essential for brain function and homeostasis during development and in the adult brain [1,2]. During CNS development, astrocytes regulate synapse formation, function, plasticity, and elimination, a function that continues in the adult and pathological brain. In the adult brain, astrocytes play multiple roles in the maintenance of brain structure and function. They provide metabolic and structural support to neurons and dynamically regulate brain homeostasis, signal transmission, and synaptic plasticity, as well as blood–brain barrier (BBB) integrity [1,2,6].

In addition to their physiological roles, astrocytes play important roles in response to disease and injury [1,2,5,6,32]. Astrocytes react to pathological insults by becoming reactive, and undergo a broad spectrum of morphological, molecular, and functional changes in a graded manner [1,2]. The changes that reactive astrocytes undergo include the downregulation of inwardly rectifying K+ (Kir) channels, the glutamate transporters GLT-1 and GLAST, the purinergic receptor P2Y1, adenosine kinase, and water transporter aquaporin 4 (AQP4); and the upregulation of metabotropic glutamate receptors [33]. These changes may cause glutamate and potassium imbalances, leading to neurotoxicity or epileptiform activity, but they do not always occur concurrently [33]. In the literature, many names have been used to describe this phenomenon. A recent recommendation suggests using “reactive astrogliosis,” “reactive astrocytes,” or “astrocyte reactivity” [1], all of which will be used interchangeably in this review.

Reactive astrogliosis is considered a defense response that aims to limit damage, control inflammation, and restore homeostasis [2,34]. However, like peripheral inflammatory responses, astrogliosis may become maladaptive and cause tissue damage under certain circumstances [33]. A recent transcriptomic analysis has revealed a high heterogeneity of reactive astrocytes and indicated that there may be spatial and temporal diversity in their response relative to the degree and site of injury [33]. The heterogeneity is thought to have functional implications, but, currently, how the heterogeneity is regulated and contributes to the adaptive and maladaptive effects of reactive astrocytes remains elusive. Efforts on identifying and categorizing beneficial and detrimental aspects of astrocyte reactivity led to the discovery of “A1” (neurotoxic and proinflammatory) and “A2” (neuroprotective) astrocytes [35,36], analogous to proinflammatory M1 and anti-inflammatory M2 macrophages. However, recent single-cell RNA sequencing (scRNA-seq) studies have demonstrated that the heterogeneity of reactive astrocytes extends beyond these two distinct states [37] and that the binary categorization of reactive astrocytes bears significant shortcomings [1]. Therefore, in the recent consensus statement, it is recommended to abandon the A1/A2 labels and the misuse of their marker genes [1]. Nevertheless, it is evident that, under certain pathological conditions, astrocytes can adopt an inflammatory phenotype and exert harmful and maladaptive effects, and markers for inflammatory reactivity are becoming established [37]. Accordingly, “inflammatory astrocytes” [37,38] or “neuroinflammatory astrocytes” [39,40] has been used to describe this type of astrocyte reactivity. “Neurotoxic reactive astrocytes” has also been used [35,41], but it is recommended to use it only when the observed neuronal death is due to identified toxic factors released by reactive astrocytes [1].

The complement system is a powerful modulator and effector of astrocyte function [42]. The activation of the complement pathway is a well-established feature of inflammatory astrocytes and can result in detrimental neuroinflammation [37]. Astrocytes are a major source of the complement system proteins [42], and the third complement component (C3) is particularly enriched in inflammatory astrocytes [37,42]. C3 mediates astrocyte toxicity to neurons, microglia, and endothelial cells [43,44,45,46]. A more recent study shows that C3+ astrocytes release long-chain saturated lipids that mediate astrocyte toxicity and induce cell death via apoptosis-related pathways [41]. Therefore, C3 exerts toxicity in many cell types via a variety of mechanisms and thus is now frequently used to identify inflammatory astrocytes [37]. Of note, currently, there is no specific marker for inflammatory reactivity and current markers for reactive astrocytes can be expressed by other cell types [37,42]. It is therefore recommended to include multiple molecular markers, together with functional assessments, when characterizing reactive astrocytes [1,2].

Inflammatory (reactive) astrocytes can be induced by microglia [35,41,47] and endothelial cells [48]. The transformation of normal astrocytes to reactive phenotypes involves a variety of intrinsic and extrinsic molecular regulators and signaling pathways, including cytokines (IL-6, LIF, CNTF, IL-1, IL-10, TGF-β, TNF-α, and INF-γ) and transcription factors (NF-κB, Stat3, Olig2, mTOR, and AP-1) [6,15,49] (Figure 2). Among them, TGF-β signaling has been identified as a key regulator of astrocyte reactivity and glial scar formation [6,15,49].

## 4. TGF-β Regulation of Astrocyte Reactivity

Astrocyte reactivity is strongly regulated by TGF-β signaling in both autocrine and paracrine fashion. Astrocytes have the ability to both synthesize and respond to TGF-β, enabling TGF-β to act as an autocrine or paracrine factor for astrocytes [14]. Astrocytes are a main contributor of endogenous TGF-β1 production in the CNS [50]. During CNS inflammation or injury, the expression of TGF-β ligands and receptors is rapidly upregulated in microglia and astrocytes, a typical hallmark of gliosis [11,51,52]. The action of TGF-β on astrocytes is highly context-dependent and can promote or inhibit astrogliosis. In primary astrocyte culture, treatment with TGF-β1 results in reactive astrogliosis characterized by the upregulation of GFAP, chondroitin sulfate proteoglycans (CSPGs), and other molecules that inhibit axon growth [53,54]. The TGF-β1-induced astrogliosis recaptures many aspects of astrogliosis and the inflammatory response in disease in vivo [55], and has thus been developed as an in vitro model of reactive astrogliosis [53]. In the presence of inflammatory stimuli, however, TGF-β1 inhibits reactive astrogliosis. Primary astrocytes cultured with IL-1α, TNF-α, and C1q show an inflammatory phenotype, which is reversed by TGF-β1 treatment [35]. The GFAP-TGF-β1 mice, which overexpress TGF-β1 specifically in astrocytes [56,57], provide an excellent tool for studying the function of astrocytic TGF-β1 in vivo. It has been shown that, depending on the disease context, increasing TGF-β1 in astrocytes in these mice can promote or inhibit astrogliosis [22,58,59].

There are several possible mechanisms by which latent TGF-β is activated in astrocytes. Many of the known activators of latent TGF-β (Figure 1) [18] have been demonstrated to function in astrocytes and activate latent TGF-β in a context-dependent manner. For example, α_v_β_8_ is enriched in astrocytes. α_v_β_8_-activated TGF-β mediates cell–cell contact and communication during development and disease [60]. The α_v_β_8_-mediated activation of TGF-β is the major mechanism of activating TGF-β in primary astrocytes and is sufficient for inhibiting endothelial migration [61]. The selective ablation of α_v_ or β_8_ in astrocytes leads to the diminished activation of latent TGF-β and defective TGF-β signaling in vascular endothelial cells, accompanied by impaired blood vessel sprouting and hemorrhage [62]. We have shown that TSP-1 is upregulated in astrocytes during neuroinflammation, leading to increased TGF-β signaling during the early stages of disease [63]. Under conditions where BBB integrity is impaired, blood constituents, such as albumin, fibrinogen, and immunoglobulins, can enter the parenchyma, accumulate particularly in astrocytes, and activate astrocytic TGF-β signaling [64,65,66].

Similarly, TGF-β may regulate astrocyte reactivity via different mechanisms. TGF-β can directly activate the GFAP promoter [14], which increases GFAP expression in astrocytes [67]. In addition, TGF-β activates many of the known molecular triggers and modulators (Figure 2) of reactive astrogliosis, including IL-6 [68], NF-κB, Jak-Stat, MAPKKK, and the complement signaling pathways [69]. TGF-β induces the rapid activation of mTOR signaling through noncanonical (SMAD-independent) pathways [70]. Astrocytes protect neurons from serum deprivation-induced cell death by the release of TGF-β and activation of the c-Jun/AP-1 signaling pathway [71]. Finally, TGF-β may modulate reactive astrogliosis through its downstream effectors. KCa3.1 (a potassium channel protein) [72], repulsive guidance molecule a (RGMa) [54], response gene to complement 32 (RGC-32) [73], and connective tissue growth factors (CTGF) [55] are all known downstream targets and have been shown to mediate TGF-β-induced reactive astrogliosis and glial scar formation. Interestingly, the effects of CTGF and TGF-β are mediated by NF-κB and AP-1 through ASK1-p38/JNK pathways [55], implicating the extensive crosstalk among astrogliosis regulators. ASK1 (apoptosis signal-regulating kinase 1) is an important mediator of astrocyte reactivity and scar formation [74]. Other well-known downstream targets of TGF-β signaling, such as Cdkn1a (p21) and plasminogen activator inhibitor-1 (PAI-1), have not been shown to be involved in TGF-β-induced astrocyte reactivity. p21 can be induced by TGF-β independent of p53 [75]. p21 plays a critical role in the mediation of TGF-β downstream effects and in the control of the specificity of TGF-β responses [76]. p21 mediates the effects of TGF-β1 on cell proliferation and can exert neuroprotective effects via cell-cycle-independent pathways through noncanonical action [77]. An absence of p21 reduces lipopolysaccharide (LPS)-induced astrogliosis [78], but whether p21 mediates TGF-β-induced astrocyte reactivity remains to be investigated. Similarly, the Smad3-dependent induction of PAI-1 in astrocytes mediates the neuroprotective activity of TGF-β1 against NMDA-induced toxicity [79], but its role in TGF-β-induced astrocyte reactivity is not yet known. In summary, the precise signaling mechanisms by which TGF-β regulates astrocyte reactivity are not fully clear, but it is likely that multiple mechanisms are involved, in a disease- and context-dependent manner.

Whether TGF-β plays a role in the regulation of inflammatory marker C3 in astrocytes is not known; however, the crosstalk between the TGF-β signaling and complement system has been shown in many other cell types, including immune cells. In human whole blood cells, Toll-like receptor 9 (TLR9) activation stimulates C3 expression by inducing TGF-β1 production [80]. Alk5 inhibitor SB431542 abolishes TLR9 stimulation on C3 gene expression [80]. TGF-β signaling is responsible for iron-induced C3 expression in the retinal pigment epithelium [81]. The pharmacologic inhibition of SMAD3 phosphorylation or knockdown of SMAD3 decreases iron-induced C3 expression [81]. In pulmonary epithelium, the treatment of TGF-β1 causes a loss of complement inhibitory proteins, leading to complement activation, whereas treatment with C3 upregulates TGF-β1 transcripts and down-regulates SMAD7 (negative regulator of TGF-β signaling) [82]. C3 inducing factors IL-1α and TNF-α upregulate TGF-β1 in astrocytes [83,84], and pretreatment with morphine suppresses the TNF-α induction of C3 and TGF-β1 [83]. Importantly, scRNA-seq analysis identified one inflammatory astrocyte subpopulation (characterized by the upregulation of C3) enriched in TGF-β signaling in Alzheimer’s disease brains [85]. These studies indicate a link between TGF-β1 and C3 induction, but whether TGF-β1 directly upregulates C3 has not been demonstrated in astrocytes.

Recent studies have suggested that TGF-β signaling may regulate astrocyte heterogeneity. During brain development, TGF-β signaling is a hallmark of astrocyte lineage diversity, and the response to TGF-β signaling drives astrocyte progenitors to generate different astrocyte lineages [86]. In the adult brain, astrocytes from different brain regions display different synaptogenic properties [87], which is, in part, due to their differential expression of TGF-β1 [88]. An in vitro study using human-induced pluripotent stem cell (iPSC)-derived astrocytes further demonstrated that the regional heterogeneity of astrocyte function is due to differences in TGF-β2 secretion between astrocytes from different regions [89]. A scRNA-seq analysis of diencephalic astrocytes revealed that astrocyte heterogeneity and low-level astrogenesis are regulated by Smad4 [90]. Likewise, the identification of an inflammatory astrocyte subpopulation enriched in TGF-β signaling in Alzheimer’s disease brains [85] suggests that the response to TGF-β signaling may drive astrocytes towards the inflammatory phenotype. As discussed above, TGF-β1 can either promote or inhibit inflammatory reactivity in vitro [35,53], depending on experimental conditions. Together, these findings point towards important roles for TGF-β signaling in astrocyte heterogeneity in the normal and diseased brain.

## 5. Astrocytic TGF-β Signaling in CNS Diseases

Reactive astrogliosis is observed in virtually all neurological conditions, including traumatic injury, stroke, aging, neurodegeneration, and epilepsy [91], as discussed in the following sections. TGF-β signaling is a powerful regulator and effector of astrocyte reactivity. The manipulation of this pathway in models of CNS injury and disease alters pathological and functional outcomes, highlighting the importance of TGF-β signaling and reactive astrogliosis in CNS diseases.

### 5.1. Traumatic Brain Injury

Traumatic brain injury (TBI) is a leading cause of death and disability and represents a significant socioeconomic and public health burden worldwide [92]. Clinically, TBI can be categorized as mild (mTBI, also referred to as concussions), moderate, or severe based on the extent of damage to the brain. The damage can be focal (confined to one area of the brain) or diffuse (widespread). Moderate and severe TBI can lead to long-term physical, cognitive, emotional, and behavioral deficits. mTBI accounts for approximately 80% of all clinically diagnosed cases and is predominantly diffuse in nature. Most patients sustaining a single mTBI recover to full function without long-term neurological impairments [93,94]. However, patients with repetitive mTBI may experience long-lasting neurological symptoms because even mild recurrent brain injuries may induce cumulative effects and interfere with neuropsychological recovery [95]. TBI has been identified as a risk factor for neurodegenerative disorders, including Alzheimer’s disease (AD) and Parkinson’s disease (PD) [96]. Animal models have been essential for our understanding of the pathophysiology and cellular and molecular mechanisms of TBI. Commonly used animal models of TBI include fluid percussion injury, cortical impact injury, weight drop-impact acceleration injury, and blast injury for non-penetrating injury [97,98]; and stab lesion and penetrating ballistic-like brain injury [99] for penetrating head injury.

The underlying pathophysiology of TBI is complex and, in general, is divided roughly into primary and secondary injury [100,101]. The primary injury is severity-dependent and results from mechanical forces applied to the skull and brain at the time of impact [100,101]. Studies utilizing various experimental models of TBI indicate that the primary brain injury triggers a cascade of molecular and biochemical events leading to long-lasting secondary neuronal and glial damage, including excitotoxicity, oxidative stress, neuroinflammation, brain edema, and delayed neuronal death [100,101].

TBI can quickly trigger astrocyte reactivity through the activation of mechanosensitive ion channels and receptors [102]. Astrocyte reactivity is an early secondary response to TBI [103,104]. Reactive astrogliosis is highly heterogeneous and can range from reversible alterations in gene expression and cell hypertrophy to scar formation with substantial cell proliferation and permanent structure reorganization [103]. Therefore, reactive astrocytes play multiple roles in TBI pathogenesis and are a major determinant of TBI outcomes [104]. Astrocyte reactivity contributes to initial synaptic loss and BBB breakdown, later synaptic remodeling [103,104], and aberrant neurogenesis in the hippocampus after TBI [105]. On the other hand, reactive astrocytes are the main cellular component of the glial scar, which is considered a protective mechanism to prevent the spreading of secondary damage [17,103]. The astroglial scar also releases inflammatory mediators to remove damaged tissue and to promote regeneration [17,103]. Thus, reactive astrocytes can have beneficial or detrimental effects following TBI.

Using the GFAP-luc mice, which express firefly luciferase reporter under the control of the GFAP promoter [106], we showed that astrocyte reactivity and GFAP expression (measured by the bioluminescence signal) correlate with injury intensity (controlled by the speed of the injury impactor) and can be used as a reliable surrogate biomarker for TBI [107,108]. The usefulness of GFAP and significance of astrocyte reactivity in TBI are highlighted by the fact that, in 2018, the FDA authorized a blood test of GFAP for the clinical evaluation of mTBI patients, because blood GFAP levels correlate with the clinical severity and extent of intracranial lesions following head trauma [109].

TGF-β is a major regulator of the injury response and has complex roles in TBI. Controversial results have been reported on the expression of TGF-β and the components of the pathway, as well as the effects of the modulation of TGF-β signaling in TBI. It is generally reported that, after traumatic injury, TGF-β signaling is activated and the components of TGF-β pathway are upregulated. For instance, increased levels of TGF-β are found in the brain following traumatic injuries in patients and in animal models [110]. TGF-β1, -β2, and -β3 [111], Tgfbr1 and Tgfbr2 [112], and Smads and p-Smads [113,114] are upregulated after injury and usually in astrocytes, oligodendrocytes, microglia, and neurons in a variety of TBI models. We observed that stab lesions resulted in the consistent induction of TGF-β1 and PAI-1 mRNA, as well as rapid activation of TGF-β signaling in SBE-luc mice [23]. On the contrary, multiple studies in preclinical models reported a reduced expression of TGF-β pathway components. The expression of Tgfb1 and Tgfbr1 is reduced in the rat brain after cortical contusion [114]. Similar results were obtained by employing NanoString gene expression analysis, where genes in the TGF-β signaling pathway (Tgfb1, Tgfbr1, Tgfbr2, Smad3, and Ski) were shown to be significantly downregulated in mouse microglia after cortical impact injury [115].

The modulation of TGF-β signaling in TBI has also shown both beneficial and detrimental effects. For example, the activation of the TGF-β signaling pathway restrains pro-inflammatory responses and boosts tissue reparatory responses of reactive astrocytes and microglia after stab wounds [113]. The intracerebroventricular (ICV) injection of TGF-β1 promotes functional recovery and alleviates axonal injury and neuroinflammation after weight drop-induced TBI [114]. Consistent with a protective role in TGF-β signaling, Tgfb1 knockdown worsens the neurological outcome in rats with weight drop TBI [116]. In contrast to these findings, a detrimental role in TGF-β signaling has also been shown by inhibiting TGF-β signaling. The inhibition of TGF-β signaling with SB431542 or transfection of Tgfb1 siRNA and inhibitory Smad7 has shown protective effects, diminishing neuroinflammation and apoptosis in a rat TBI model of fluid percussion injury [117]. Similarly, blocking TGF-β signaling with angiotensin receptor 2 antagonist losartan ameliorates secondary brain injury in a cortical impact injury model, decreases the brain lesion volume and neuronal apoptosis, and improves the neurological and motor function [118]. The reason for these contradictory results is not clear, but it is most likely due to the complex pathophysiology of TBI and/or differences in animal models employed, and the stages of injury and cell type affected.

The role of TGF-β signaling in astrocytes in TBI seems to be more defined. The TBI-induced astrocyte reactivity and activation of TGF-β signaling in astrocytes may be triggered by the exposure of astrocytes to blood-borne factors, such as albumin, fibrinogen, and thrombin. Albumin stimulates the secretion of TGF-β and IL-1β from astrocytes [64]. Fibrinogen is a carrier of latent TGF-β and induces p-Smad2 in astrocytes, which leads to astrocyte reactivity and the inhibition of neurite outgrowth [65]. The genetic or pharmacologic depletion of fibrinogen in mice reduces active TGF-β, p-Smad2, gliosis, and neurocan deposition after TBI [65]. The stereotactic injection of fibrinogen into the mouse cortex induces astrogliosis, and the inhibition of TGF-β signaling abolishes fibrinogen-induced glial scar formation [65]. Similarly, the ICV infusion of thrombin activates TGF-β signaling [119] and drives astrogliosis and memory impairment [120]. Thrombin-induced astrocyte dysfunction contributes to depression [121] and seizure [122] following TBI. These results identify blood-borne factors as a primary activation signal for astrocyte reactivity and TGF-β signaling, and support a detrimental role of astrocytic TGF-β signaling in TBI. They point to TGF-β signaling as a molecular link between vascular damage and astrogliosis.

### 5.2. Stroke

Stroke is the second leading cause of death and third leading cause of disability in adults worldwide [123]. Stroke can be classified into two major types: ischemic and hemorrhagic. Hemorrhagic stroke results from a ruptured blood vessel and the extravasation of blood into the brain parenchyma. Ischemic stroke accounts for approximately 80% of all strokes and results from the thromboembolic occlusion of a cerebral artery. Cerebral artery occlusion results in a decreased blood flow, which leads to neuronal dysfunction/death and clinical deficits [123]. The ischemic penumbra, the area of hypo-perfused brain tissue surrounding the ischemic core, could potentially be salvaged and is therefore the focus of stroke research and clinical practice [124]. Following ischemic stroke, the most pronounced pathological and cellular change is the glial response: microglia within and around the infarct engulf cellular debris, astrocytes around the infarct proliferate and form glial scars, and astrocytes in the ischemic penumbra become reactive in a spatial gradient [33].

Reactive astrogliosis is a prominent pathological feature after stroke [33]. The transcriptomic profile of astrocytes after ischemic injury shows a beneficial anti-inflammatory phenotype [36]. In the acute phase, astrocytes limit the tissue damage by promoting brain homeostasis. In the post-acute, recovery phase, reactive astrocytes modulate axonal sprouting and synaptic plasticity, and participate in CNS regeneration [125,126]. Accordingly, the ablation of a subset of reactive astrocytes disrupts vascular repair and remodeling, exacerbates vascular permeability, and worsens motor recovery [127]. However, the swelling endfeet of reactive astrocytes can compress brain microvessels and thereby decrease microvascular perfusion [128]. Therefore, astrogliosis and astrocyte dysfunction have been linked to post-stroke cerebral blood flow (CBF) reduction and BBB impairment [33]. Consequently, the inhibition of astrogliosis improves CBF and reduces cerebral microvessel damage and BBB injury in ischemic mouse brains [129], suggesting a detrimental role of astrogliosis after stroke. A number of key factors and intracellular signaling pathways have been discovered to govern astrocyte behavior. Among them, TGF-β is identified to support the protective phenotype of reactive astrocytes after stroke [15]. 

The neuroprotective function of TGF-β is most established in brain ischemia [130]. The mechanisms by which TGF-β mediates neuroprotection include the suppression of inflammation, apoptosis, and excitotoxicity, as well as the promotion of scar formation, angiogenesis, and regeneration [11,130]. For example, the ICV delivery of TGF-β1 reduces infarction size and suppresses neuronal apoptosis in rats after ischemic stroke [131]. TGF-β1 produced in the ischemic core is shown to diffuse toward the ischemic penumbra and drive microgliosis to eliminate degenerating neurons [132]. Accordingly, the overexpression of Smad3 in the rat brain reduces infarct volume through anti-inflammatory and anti-apoptotic pathways [133]. In summary, TGF-β exerts neuroprotective effects through multiple mechanisms, but it is not clear whether these effects are mediated through neurons or glial cells.

In the SBE-luc mice, the ischemic-stroke-induced activation of TGF-β signaling begins on day 1 and peaks on day 7 [134]. Microglia [134] and astrocytes [135] are the predominant source of TGF-β production after stroke. TGF-β signaling is activated in astrocytes and microglia in the stroke penumbra [134]. Like in TBI, TGF-β signaling in astrocytes after stroke can be activated by albumin through BBB breakdown [64]. The role of TGF-β signaling in astrocytes has been studied in the “Ast-Tbr2DN” mice, which express a dominant negative mutant form of Tgfbr2 specifically in astrocytes. These mice display exacerbated neuroinflammation and worse motor outcomes after stroke [136]. These findings are in line with recent work reporting that the blockade of Tgfbr2 in astrocytes abolishes zinc finger E-box binding homeobox 1 (ZEB1)’s protective effects against acute ischemic brain injury [137]. Together, these experiments demonstrate that TGF-β signaling in astrocytes plays a protective role in stroke. Of note, astrocytes isolated from stroke mice show neuroprotective properties by transcriptomic analysis, which has led to the identification of neuroprotective A2 astrocytes [35,36]. TGF-β1 could revert A1 astrocytes to a non-reactive phenotype [35], but whether the activation of TGF-β signaling skews astrocytes toward a protective A2 phenotype remains to be determined.

### 5.3. Aging

Astrocytes are vulnerable to age-associated dysfunction and stress [138] and undergo complex and region-specific morphological, molecular, and functional changes upon aging [138,139]. Astrocytes display a reactive phenotype with compromised homeostatic functions as the brain ages [138,140]. These include a decreased support of neurons and synapses, impaired synapse formation and synaptic transmission, and a decreased support of BBB integrity [138,140]. Transcriptomic studies have revealed that aged astrocytes display signatures indicative of inflammatory astrocyte reactivity, with highly up-regulated genes involved in the complement (*C3* and *C4b*), peptidase inhibitor (*Serpina3n*), and cytokine (*Cxcl10*) pathways [141,142,143]. Interestingly, C3 and other members of the complement cascade are strongly upregulated in aging astrocytes in all brain regions [141]. A meta-analysis study of 591 gene expression datasets from human prefrontal cortices of distinct ages revealed that the most outstanding results were the age-related decline of synaptic transmission and activated expression of GFAP in the aging brains [144]. Interestingly, the regional expression patterns of astrocyte-specific genes also change upon aging, particularly in the hippocampus and substantia nigra [145], two key areas involved in Alzheimer’s and Parkinson’s disease. In summary, these results support that, with aging, astrocytes develop a region-dependent reactive and inflammatory phenotype [138,139]. The region specificity might contribute to the differential regional vulnerability to aging and age-related neurodegenerative disorders [138]. Besides reactive astrogliosis, it has been recently proposed that asthenia with a loss of function is a main feature of astrocyte dysfunction in the aging brain [146].

Age-related alterations of the TGF-β pathway in the brain include TGF-β and Smad expression and the TGF-β-induced glial response. Whereas TGF-β levels are increased with aging in the brain of humans [147] and mice [148], Smad expression is reduced with aging. Smad2(Δexon3), a splice form of Smad2 lacking exon3, directly binds to the DNA, resulting in a functional hybrid of Smad2 and Smad3. Smad2(Δexon3) is the most abundant Smad2 isoform in the brain, and is strongly increased prenatally and in early postnatal life, but it continuously diminishes as the brain matures and ages [149]. Such an age-related reduction in Smad expression could lead to impaired TGF-β signaling. Indeed, through longitudinal in vivo bioluminescence monitoring, we recently observed that brain TGF-β signaling decreases with age in the SBE-luc mice [26]. The aged brain not only has reduced TGF-β signaling, but it also fails to upregulate TGF-β1, Smad3, and p-Smad3 upon LPS stimulation [148], suggesting an impaired response to inflammatory stimuli. The impairment of TGF-β signaling could contribute to the persistent mild neuroinflammation [150] and reduced adult neurogenesis [151] during aging. In agreement with this view, the photoactivation of the TGF-β signaling pathway promotes adult hippocampal neurogenesis [152].

The age-related changes in TGF-β signaling are cell-type-dependent. TGF-β1 is reduced in aged neurons [28]. The expression of TGF-β1 is also reduced in oligodendrocytes with aging, which is thought to impair the differentiation of oligodendrocyte progenitor cells into myelinating oligodendrocytes [153]. In contrast, TGF-β1 is increased with aging in astrocytes [154]. In vitro, astrocytes from old animals consistently secrete higher amounts of TGF-β in vitro compared with the cells from postnatal or young animals [155]. Collectively, these studies suggest that the increased TGF-β observed in the aged brain may be predominantly from astrocytes. The activation of TGF-β signaling in astrocytes has been linked to BBB breakdown in aged human and rodent brains. BBB breakdown is an early biomarker of human cognitive dysfunction [156]. When BBB integrity is compromised, blood-borne factors such as albumin, fibrinogen, and immunoglobulins accumulate in aging human and rodent brains, particularly in astrocytes [157]. These blood-borne factors trigger the activation of TGF-β signaling in astrocytes, which is necessary and sufficient to cause neuronal dysfunction and age-related pathology in rodents [66]. The toxic effects of albumin extravasation have been demonstrated by showing that a direct infusion of albumin into the young mouse brain induces astrocytic TGF-β signaling and an aged brain phenotype [66]. In addition, the conditional genetic knockdown of Tgfbr2 in astrocytes or pharmacological inhibition of Tgfbr1 attenuates age-related cognitive decline and vulnerability to seizures in mice [66]. These observations establish TGF-β as a novel link between reactive astrocytes and cognitive decline. In summary, TGF-β signaling is activated in astrocytes during aging and promotes age-related cognitive deficits. The inhibition of TGF-β signaling may offer therapeutic benefits against cognitive impairments during aging.

### 5.4. Alzheimer’s Disease

Alzheimer’s disease (AD) is the most common form of dementia and one of the top global health concerns [158]. Clinically, AD is characterized by progressive cognitive impairments, mainly learning deficits and memory loss [158]. Pathologically, AD is characterized by extracellular amyloid plaques of amyloid β (Aβ) and intracellular neurofibrillary tangles (NFTs) of hyperphosphorylated tau (p-tau), accompanied by reactive gliosis [158]. AD risk genes are mainly expressed by glial cells [3]. For example, apolipoprotein E (APOE), clusterin (CLU), and fermitin family member 2 (FERMT2) are predominantly expressed by astrocytes, supporting a critical role of astrocytes in AD pathophysiology [159].

Reactive astrogliosis is a common and widespread pathological feature of AD brains [8,49,160]. Reactive astrocytes are detected during the early phases of AD, before the presence of characteristic AD pathology, and their reactivity increases with disease progression [3,8]. Recent studies demonstrate that there may be a causal relationship between reactive astrocytes and neurodegeneration. In an animal model where astrocyte reactivity is finely controlled, mild reactivity naturally reverses its reactivity, whereas severe reactivity causes irreversible neurodegeneration and cognitive deficits [161]. The severe reactivity also induces neurodegeneration in APP/PS1 mice, an AD model known for a lack of neurodegeneration [161]. In another study, the overexpression of 3R tau, specifically in hilar astrocytes, leads to reactive astrogliosis and impaired adult neurogenesis and spatial memory performances [162]. Accordingly, the inhibition of astrocyte reactivity through the STAT3 pathway attenuates amyloid deposition and synaptic and spatial learning deficits in AD model mice [163]. These studies demonstrate that reactive astrocytes are sufficient to cause neurodegeneration and may be key players in the etiology of AD [3,5,15,164].

One potential mechanism in which reactive/inflammatory astrocytes contribute to neurodegeneration is through C3 [35]. The upregulation of C3 is observed in AD astrocytes from bulk RNA-seq [165] and snRNA-seq [85] analyses. In the cortex of AD patients, around 60% of the astrocytes are C3 + [35]. Exposure to Aβ activates NF-κB and C3 release in astrocytes [45]. C3 mediates communication/neurotoxicity from astrocytes to other brain cells, contributing to AD pathogenesis. C3 binding to the receptor C3aR in microglia attenuates microglial phagocytosis [44], binding to C3aR in neurons disrupts dendritic morphology and network function [45], and binding to C3aR in endothelial cells promotes vascular inflammation and BBB dysfunction [46]. Accordingly, the genetic deletion of C3 mitigates the Aβ and tau pathology, neurodegeneration, and functional deficits in AD models [44,45,166,167,168]. Together, reactive astrocytes contribute to maladaptive effects in part through proinflammatory reactions, where C3 acts as a perpetrator of neuroinflammation and neurotoxicity.

An early microarray and RNA-seq analysis of acutely isolated astrocytes from APP mouse models [169] and astrocytes microdissected from AD patients [165] revealed that astrocytes acquired an inflammatory phenotype, with less supportive capacity to neurons. Recent scRNA-seq/snRNA-seq studies of human AD brains have revealed a high heterogeneity and identified many astrocyte subpopulations/clusters. These AD pathology-associated astrocyte subpopulations display different gene expression signatures characterized by, for example, (1) the upregulation of GLUL and CLU, and downregulation of APOE [170]; (2) enrichment for TGFβ signaling and immune responses, with upregulation of C3 [85]; (3) downregulation of genes implicated in metabolic coordination [171]; (4) expression of Aβ plaque-induced genes [172]; (5) enriched expression of stress response-associated genes [173]; (6) GFAP-high subpopulation with upregulation of genes involved in the extracellular matrix and proteostasis [174]. In the 5xFAD transgenic mouse model of AD, astrocytes undergo dynamic responses as the disease progresses, from a GFAP-low to a GFAP-high state, and an AD-specific population (termed “disease-associated astrocytes”) [175]. These “disease-associated astrocytes” are enriched in Gfap, Serpina3n, Ctsb, ApoE, and Clu. Overall, these studies provide complementary snapshots of astrocytic responses to AD pathology [159]. They suggest that reactive astrocytes in AD are highly heterogenous and may contribute to different aspects of AD pathology. A recent systemic review of 306 publications further supports this notion, showing that AD reactive astrocytes undergo a wide range of functional changes [176]. Interestingly, 3 of the 196 AD astrocyte proteins, TGFB2, TGFB3, and TGFBR2 [176], are related to the TGF-β pathway, highlighting the significance of TGF-β signaling in AD pathology.

TGF-β signaling has long been implicated in the pathogenesis of AD. A genetic polymorphism in TGFB1 is associated with the risk of developing AD [177]. TGF-β1 levels are increased in post-mortem AD brains [178]. However, the brains of AD patients have reduced levels of TGFBR2 [179], as well as decreased nuclear Smad2, Smad3, and Smad4 [180]. Nuclear p-Smad2 and p-Smad3 are also reduced in neurons [181,182,183]. Since the nuclear translocation of p-Smad2/3 is required for the activation of TGF-β signaling and transcription of TGF-β target genes (Figure 1), these observations suggest a defect of TGF-β/Smad signaling in neurons in AD. Such a defect is considered to compromise the neuroprotective effects of TGF-β/Smad signaling, as reducing TGF-β signaling in neurons resulted in age-dependent neurodegeneration and promoted Aβ accumulation and dendritic loss in a mouse model of AD [179]. It has therefore been proposed that impaired TGF-β signaling in neurons contributes to Aβ accumulation and neurodegeneration, and is a risk factor for AD [184].

On the contrary, TGF-β signaling in astrocytes seems to be detrimental. The enrichment of TGF-β signaling and upregulation of reactivity marker gene C3 in an AD-associated astrocyte subpopulation [85] indicate that TGF-β signaling may promote inflammatory astrocyte reactivity. The astrocyte-targeted overexpression of TGF-β1 promotes amyloid angiopathy in the frontal cortex and meninges [56,57] and increases the production of Aβ40/42 by astrocytes in GFAP-TGF-β1/APP (amyloid precursor protein) transgenic mice [59]. Interestingly, TGF-β1 drives APP production only in astrocytes and not in neurons [59], suggesting an astrocyte-specific mechanism of TGF-β contributing to AD pathology. More strikingly, the GFAP-TGF-β1 mice (without overexpressing mutant APP) develop an AD-like cerebrovascular pathology, including a reduction in CBF and increase in perivascular Aβ accumulation [58], further supporting the idea that astrocyte TGF-β1 can directly induce an AD-like vascular and amyloid pathology. Astrocyte TGF-β signaling also mediates APOE neurotoxicity and contributes to the risk of AD. A whole-body or astrocyte-specific deletion of ApoE significantly ameliorates spatial learning and memory deficits, reduces Aβ production, and inhibits astrogliosis in APP transgenic mice [185]. The overexpression of TGF-β in astrocytes abrogates the protective effects of ApoE knockout. In contrast, the inhibition of TGF-β in astrocytes of APP mice exerts therapeutic effects similar to ApoE knockout [185]. In summary, it seems like astrocytic TGF-β signaling plays a detrimental role in AD.

### 5.5. Parkinson’s Disease

Parkinson’s disease (PD) is the second most common neurodegenerative disease after AD [4,186,187]. Neuropathological hallmarks of PD are a loss of dopaminergic (DA) neurons in the substantia nigra pars compacta (SNpc) and intracellular aggregates of α-synuclein (α-syn) [4,186,187]. A loss of DA neurons causes striatal dopamine deficiency, which has been identified as the main cause of the disease’s movement symptoms [4,186,187]. The molecular mechanism of neurodegeneration in PD remains largely unknown, but likely involves multiple pathways and cell types. Neuroinflammation in the SNpc, including astrocyte reactivity, is a key feature of PD pathophysiology [4,186,187]. Accumulating evidence suggests that reactive astrocytes have a crucial role in initiating PD pathophysiology, rather than being merely a secondary phenomenon in response to the damage and death of DA neurons [4].

Genome-wide association studies have revealed numerous loci associated with the risk of development of PD [188]. Many of the genes identified in these studies are expressed in astrocytes at similar or greater levels than in neurons [189], supporting an important role of astrocytes in PD pathogenesis [4]. A systemic analysis indeed revealed that PD-associated genes are enriched in astrocytes of the cortex and substantia nigra [190]. The enrichment of PD heritability from PD GWAS datasets is observed in a lysosomal-related gene set that is highly expressed in astrocytes, microglia, and oligodendrocyte subpopulations [191]. Importantly, genes known to be causative in PD have important roles in astrocyte function [4]. For example, PARK7 (DJ-1), SNCA (α-syn), PLA2G6 (iPLA2), ATP13A2, PINK1, and PRKN (Parkin) are all involved in astrocyte-specific functions, including inflammatory responses, glutamate transport, and neurotrophic capacity [4]. Recent snRNA-seq studies have revealed neuroinflammatory signatures in astrocytes of idiopathic PD patients [192] and α-syn-A53T mice [193], supporting a pro-inflammatory and disease-promoting role of reactive astrocytes. The inflammatory (C3+) astrocyte phenotype is induced by 1-methyl-4-phenyl-1,2,3,6-tetrahydropyridine (MPTP) [194] or β-sitosterol-β-D-glucoside (BSSG) [195], neurotoxins used to model PD. The genetic deletion of the potassium channel subunit Kir6.2 [196] or treatment with NLY01, a glucagon-peptide-1 receptor agonist [197], mitigates inflammatory astrocyte reactivity and prevents dopaminergic neurodegeneration and behavioral deficits. A direct demonstration of non-cell autonomous mechanisms during neurodegeneration was reported in a mouse model, where PD-related A53T α-syn mutation is specifically overexpressed in astrocytes [198]. These mice show widespread astrogliosis, dopaminergic neuronal loss, and movement disabilities [198], suggesting that α-syn in astrocytes is sufficient to impair astrocyte function and initiate neurodegeneration. In a *Drosophila* model of PD, where α-Syn and risk gene expression are manipulated in neurons and glia separately, several glial risk factors that modify neuronal α-Syn toxicity are identified [199]. The knockdown of these genes exacerbates dopaminergic neuron loss and increases α-syn oligomerization [199]. These results suggest that the PD risk genes exert their effects in glia, which influence neuronal α-Syn proteostasis in a non-cell-autonomous manner [199].

TGF-β has multiple associations with the nigrostriatal system and with pathological characteristics of PD, including DA neuron development and survival, dopaminergic degeneration, α-syn aggregation, and γ-Aminobutiryc acid (GABA) neurotransmission [200,201]. TGF-β is essential for the development and survival of embryonic DA neurons [201], promoting the survival of DA neurons in culture and protecting them against toxicity from the Parkinsonism-inducing toxin N-methyl pyridinium ion (MPP^+^, a neurotoxic metabolite of MPTP) [202]. Genetic association studies suggest that a variation in the TGFB2 gene may influence susceptibility to idiopathic PD [203]. TGF-β2 haplodeficiency (Tgfb2^+/-^) mice have fewer DA neurons in the SN and a significantly reduced dopamine concentration in the striatum in adulthood [204]. Smad3 and pSmad3 expression decreases with age in mouse substantia nigra [205]. Smad3-deficient mice develop a progressive loss of DA neurons and aggregation of α-synuclein [206]. These results suggest that deficiency in TGF-β signaling may increase the risk of developing PD. However, the continuous administration [207] or overexpression [208] of TGF-β fails to show protective effects against MPTP in animal models of PD. The cause of the contradictory results from in vitro and animal studies is not known, but could be that the in vivo delivery of the TGF-β ligands may initiate TGF-β signaling in many cell types in vivo [209]. It is therefore important to study TGF-β signaling in a cell-type-specific manner. We used transgenic mice and viral-mediated gene transfer to drive the expression of mutant TGF-β receptors and achieve a neuron-specific manipulation of TGF-β signaling [209]. We generated mice with reduced TGF-β signaling in neurons by expressing a truncated kinase-defective Tgfbr2 under the control of a CamKII promoter [209]. These mice display age-related motor deficits and a mild degeneration of midbrain dopaminergic neurons [209]. Similarly, deleting Tgfbr2 in mature dopaminergic neurons with DAT-iCre caused a significant reduction in dopaminergic axons in the striatum in another study [210]. Moreover, we show that increasing TGF-β signaling by the overexpression of a constitutively active form of Tgfbr1 reduces MPTP-induced dopaminergic neurodegeneration and motor deficits [209]. In agreement with this finding, we recently show that C381 (formerly SRI-011381), a novel small molecule TGF-β/Smad activator, significantly reduces MPTP-induced dopaminergic neurodegeneration and improves motor function in mice [26]. Together, these studies suggest a protective role of neuronal TGF-β signaling in PD.

Compared with neuronal TGF-β signaling, astrocytic TGF-β signaling in PD is less studied. TGF-β signaling may be compromised in PD astrocytes, as the RNA sequencing of LRRK2 G2019S iPSC-derived astrocytes revealed a downregulation of TGFB1 [211]. The ICV injection of α-syn oligomers increases TGF-β1 secretion by reactive astrocytes [212,213]. The inhibition of TGF-β signaling reduces the density of striatal excitatory synapses and the expression of astrocyte glutamate transporters [212,213], supporting a protective role of TGF-β signaling in synucleinopathy. Astrocytes of aquaporin 4-deficient (Aqp4^-/-^) mice fail to upregulate TGF-β1 in response to MPTP treatment and display significantly stronger inflammatory responses and greater losses of dopaminergic neurons than wild-type controls [50]. The stereotactic injection of TGF-β1 in the striatum significantly reduces neuronal damage and microglial activation in MPTP-treated Aqp4^-/-^ mice [50]. These findings support the idea that astrocytic TGF-β1 is a potent inhibitor of neuroinflammation and mitigates dopaminergic neuron injury, and that AQP4 regulates the TGF-β pathway in astrocytes.

In addition to PD, atypical Parkinson disorders also exhibit parkinsonism, but with different clinical manifestations and pathologic features to PD [214]. Atypical parkinsonian disorders commonly include dementia with Lewy bodies (DLB), multiple system atrophy (MSA), progressive supranuclear palsy (PSP), and corticobasal degeneration (CBD) [214]. The pathogenesis of atypical Parkinson disorders is not well understood, but current evidence suggests that the inflammatory mechanisms in PD and atypical parkinsonisms appear to differ [215]. Pathologically, PSP and CBD are tauopathies. PSP is characterized by tau-enriched tufted astrocytes and NFTs in subcortical nuclei [214,216]. The pathologic features for CBD are cortical and striatal tau-positive neuronal and glial lesions, especially astrocytic plaques and thread-like lesions, along with neuronal loss in focal cortical regions and in the substantia nigra [214,216]. The tau pathology in astrocytes does not correlate with neuron loss and is thus considered as an independent degenerative process rather than a reactive response [215]. It may be noted that PSP and CBD share multiple features in the clinical manifestations, pathology, biochemistry, and genetic risk factors [217,218,219]. The boundaries between PSP and CBD are thus often questionable and, currently, there is no definitive noninvasive antemortem diagnostic test [217,220]. DLB and MSA are synucleinopathies. Though α-syn is expressed predominantly in neurons, α-syn aggregates and inclusions in astrocytes are a common feature in these neurodegenerative diseases [221]. Astrocytes have been shown to interact with extracellular α-syn released by neurons and mediate neuroinflammation, cell-to-cell spread, and other aspects of pathogenesis [222]. Together, these pathological features indicate that astrocytes may be a key element in the pathogenesis of atypical Parkinson disorders [223]. The expression of TGF-β2 is observed in both NFT-bearing neurons and tangle-bearing glial cells, and the immunoreactivity of TGFBR1 and TGFBR2 is increased in reactive glia in PSP [224,225], but the role of TGF-β has not been investigated.

### 5.6. Amyotrophic Lateral Sclerosis

Amyotrophic lateral sclerosis (ALS or Lou Gehrig’s disease) is a devastating neurodegenerative disease characterized by a progressive loss of motor neurons, with an incidence of 1–2/100,000 per year and mean survival of 3–5 years after diagnosis [226]. ALS occurs in two different forms, sporadic (>90% of cases) and familial (<10% of cases). To date, more than 120 genetic variants have been implicated in ALS (https://alsod.ac.uk accessed on 18 April 2022), and at least 25 genes have been shown to cause or significantly increase the risk of ALS. Among them, C9ORF72, SOD1, FUS, and TARDBP (TDP-43) are the most common causative genes [7,226]. The precise pathological mechanisms of ALS are not clear. Besides the main classical “neuron-centric” view, recent studies have substantiated that ALS could also be a non-cell-autonomous disease [7]. Gliosis is a pathological hallmark of ALS, and glial cells are all able to modulate the ALS pathology [7].

Experimental models of ALS have greatly improved our understanding of reactive astrocytes in ALS pathogenesis [7]. Most studies across ALS model systems support a neurotoxic role of reactive astrocytes in ALS [7]. C3^+^ inflammatory astrocytes are abundant in familial and sporadic ALS patients and in *SOD1^G93A^* mice [39]. Knocking out C3-inducing factors (IL-1α, TNF-α, and C1q) markedly extends survival in the *SOD1^G93A^* ALS mouse model [39]. Direct evidence of a non-cell autonomous toxic mechanism came from models using cell-type-specific promoters. Whereas the selective deletion of SOD1^G85R^ [227] or SOD1^G37R^ [228] in astrocytes slows disease progression, the astrocyte-restricted expression of TDP43^M337V^ leads to astrogliosis and progressive non-cell-autonomous motor neuron loss and paralysis [229]. Transplantation studies further support the neurotoxicity of reactive astrocytes. The transplantation of SOD1^G93A^ astrocyte precursors induces host motor neuron death and motor dysfunction in wild-type rats [230], whereas the transplantation of wild-type glial precursors that have differentiated into astrocytes in the spinal cord of SOD1^G93A^ rats prolongs survival and attenuates motor neuron loss and forelimb motor deficits [231]. In vitro, iPSC-derived astrocytes from patients with VCP and SOD1 mutations undergo cell-autonomous reactive transformation characterized by an increased expression of C3 [232]. Accordingly, astrocytes derived from adult NPCs isolated from post-mortem sporadic or familial ALS patients are toxic to healthy motor neurons in culture [233,234]. The neurotoxicity of reactive astrocytes seems to be mediated by neurotoxic factors and the failure of astrocytes to support motor neurons [7]. TGF-β is one of the neurotoxic factors mediating the neurotoxicity of ALS astrocytes [7,235].

A meta-analysis of human and mouse multi-omics reveals that ALS astrocytes are characterized by signatures of TGF-β signaling and inflammatory reactivity, i.e., the upregulation of genes involved in the extracellular matrix, endoplasmic reticulum stress and the immune response, and downregulation of neuronal support processes [236]. These features suggest that ALS astrocytes are pro-inflammatory and lose protective functions. Studies from both cell culture and animal models demonstrate that TGF-β signaling in ALS astrocytes plays a detrimental role and contributes to ALS pathology. In a co-culture model of human motor neurons and primary astrocytes, TGF-β1 secreted by reactive astrocytes disrupts autophagy and induces protein aggregation in motor neurons [237]. In the spinal cord of SOD1^G93A^ mice, TGF-β1 is upregulated mainly in astrocytes, and Tgfb1 mRNA levels negatively correlate with the mouse lifespan [238]. The astrocyte-specific overproduction of TGF-β1 accelerates disease progression in the GFAP-TGF-β1/SOD1^G93A^ mice [238]. Furthermore, the inhibition of TGF-β signaling by a selective Tgfbr1 inhibitor, SB-431542, extends the survival time of SOD1^G93A^ mice [238]. In summary, these studies show that astrocyte TGF-β1 promotes the ALS pathology, and that astrocytes adopt neurotoxic properties in ALS.

### 5.7. Multiple Sclerosis

Multiple sclerosis (MS) is a chronic inflammatory and degenerative disease of the CNS characterized by focal neuroinflammatory lesions and demyelination [239,240]. MS is one of the most prevalent neurological disorders among young adults, with most cases diagnosed between 20 and 50 years of age [239,240]. MS patients show a variety of physical disabilities and cognitive impairments, with different disease progression patterns. Based on disease progression patterns, four types of MS have been identified: relapsing-remitting (RR), primary progressive (PP), secondary progressive (SP), and clinically isolated syndrome (CIS) [241]. Relapsing-remitting MS (RRMS) is the most common type, accounting for more than 80% of MS cases, and many RRMS further develop into secondary progressive MS (SPMS) [241,242].

MS is an autoimmune disease in which the immune system attacks the myelin sheath covering the axons, leading to inflammatory demyelinating lesions [240,243]. The immune attack is mediated by autoreactive T-cells, T-helper (Th)-1, and Th-17 [240,243]. Other immune cells, including cytotoxic T-cells, B-cells, monocytes/macrophages, and CNS glial cells, namely astrocytes and microglia, are also involved in the inflammatory attack and inflict myelin damage [240,243]. MS lesions can be classified into (i) acute lesion with numerous inflammatory cells and astroglial hypertrophy, (ii) chronic active lesion with edged demyelination, and (iii) chronic lesion with fewer leukocytes but profound demyelination, axonal loss, and astrogliosis [241] (based on [244]). Thus, astrogliosis is a key component of MS lesions, and reactive astrocytes play a critical role in lesion development [5].

The roles of astrocytes in MS are complex, and both beneficial and harmful roles have been attributed to reactive astrogliosis [242,245,246,247]. In both MS and its animal model, experimental autoimmune encephalomyelitis (EAE), astrocyte reactivity initiates during the early stage of lesion formation and persists into the chronic phases, even after the immune cell presence has receded [242,246,248]. C3-containing inflammatory astrocytes are abundant in MS lesions [35], as well as in EAE mice [249]. The conditional deletion of C3 in astrocytes attenuates EAE-induced axonal injury [250]. Using the GFAP-luc mice, we observed that astrocyte reactivity in EAE starts several days before the onset of paralysis, correlating with and predicting EAE clinical severity [22,251]. These findings suggest that reactive astrocytes promote inflammation and lesion formation and that astroglial reactivity is a reliable indicator of disease evolution [246,248]. In agreement with this view, the inhibition of astrocyte reactivity suppressed local CNS inflammation and neurodegeneration in EAE [252]. The underlying mechanisms of reactive astrocytes promoting inflammatory lesion include the recruitment of peripheral inflammatory cells and the activation of microglia and astrocyte intrinsic neurotoxic activities [5,32]. On the other hand, it has been shown that astrocytes have beneficial effects by restricting the infiltration of peripheral immune cells into the CNS and releasing neurotrophic factors to promote tissue repair [5,32]. The transcriptomic profile of astrocytes isolated from non-demyelinated normal-appearing white matter supports a neuroprotective role of reactive astrocytes [253].

The complex roles of astrogliosis in MS and EAE may be explained by the heterogeneity of reactive astrocytes [247], as demonstrated by recent transcriptomic studies. Astrocyte reactivity has been shown to differ between CNS regions in EAE [254] and in MS lesions [255]. Astrocyte heterogeneity is further demonstrated by the discovery of novel astrocyte subsets from recent scRNA-seq studies [256,257,258]. For example, a pro-inflammatory and neurotoxic astrocyte subset characterized by the downregulation of NRF2 and upregulation of MAFG has been discovered in EAE and MS [258]. NRF2 is a negative regulator of inflammation and oxidative stress. MAFG decreases NRF2 and interacts with MAT2α to block anti-inflammatory pathways. The downregulation of NRF2 and upregulation of MAFG likely promote inflammatory responses in EAE [258]. Similarly, a novel astroglial phenotype with neurodegenerative programming, “astrocytes inflamed in MS,” is identified in another study using MRI-informed snRNA-seq [256]. Interestingly, an anti-inflammatory subset of astrocytes characterized by the co-expression of LAMP1 and TRAIL has also been identified in EAE and MS [257]. This astrocyte population limits inflammation by inducing T-cell apoptosis and is driven by IFNγ [257]. Collectively, these studies suggest that reactive astrocytes display functional and phenotypic heterogeneity and can adopt either a pro-inflammatory or an anti-inflammatory phenotype under CNS autoimmune conditions [5]. A balance between the pro-inflammatory and anti-inflammatory subsets may be critical for the onset and progression of EAE and MS.

Given the potent immunomodulatory effects of TGF-β and the autoimmune etiology of MS and EAE, it is not surprising that TGF-β signaling has been extensively studied in the context of MS and EAE. TGF-β controls both innate and adaptive immune responses by regulating the generation and effector functions of many immune cell types [259]. TGF-β plays a major role in the development and function of both encephalitogenic and regulatory T-cells (Treg) [260]. It also regulates the complex behavior of natural killer cells, macrophages, and neutrophils [259]. These activities likely underlie the beneficial effects of the systemic administration of TGF-β in EAE models [261].

In both active demyelinating and chronic MS lesions, all three TGF-β isoforms and their receptors are strongly expressed in hypertrophic astrocytes [262]. It is therefore hypothesized that astrocyte TGF-β signaling participates in reactive processes and promotes the formation of chronic MS lesions. In the SBE-luc reporter mice, we observed EAE-induced early TGF-β1 production in glial cells and TGF-β signaling in the CNS several days before the onset of paralysis in EAE mice [22]. The astrocyte-targeted overexpression of TGF-β1 results in an earlier onset and more severe paralysis in the GFAP-TGFβ1 mice [22], and systemic treatment with pharmacological inhibitors of TGF-β signaling (Tgfbr1 antagonist or losartan) ameliorates the paralytic disease [22,63]. Therefore, the early production of TGF-β1 in astrocytes may create a permissive environment for the initiation of autoimmune inflammation in EAE. Recent findings from transcriptomic studies support this hypothesis. In patients with progressive MS, large areas of periplaque astrogliosis, partial demyelination, and low-grade inflammation in the spinal cord extend away from plaque borders; these areas are characterized by the up-regulation of TGF-β signaling [51,52]. Similarly, in a study of white matter from patients with progressive MS, RNA-seq analysis and de novo network enrichment based on shared DEGs discovered TGFBR2 as a central hub, which was most upregulated in remyelinating lesions [263]. RNAscope and immunohistochemistry demonstrated astrocytes as the cellular source of TGFBR2 [263]. Together, these findings support the idea that TGF-β signaling in astrocytes promotes astrogliosis, demyelination, and chronic inflammation, and that targeting TGF-β signaling might be a promising therapeutic strategy for MS.

### 5.8. Huntington’s Disease

Huntington’s disease (HD) is an autosomal dominant neurodegenerative disorder characterized by motor disabilities, cognitive impairments, and psychiatric disturbances [264,265,266]. The genetic cause of HD is an expansion of a polyglutamine-encoding CAG repeat in exon 1 of the huntingtin gene (HTT), with longer repeat lengths leading to an earlier onset and more severe disease [264]. A neuropathological analysis of post-mortem HD brains reveals a dramatic degeneration of neurons prominent in the striatum and cerebral cortex [264]. Neuroinflammation, characterized by the presence of reactive microgliosis and astrogliosis, is frequently observed in HD patients before symptom onset [267]. The classical hypothesis postulates that HD pathogenesis stems from the toxicity of mutant HTT (mHTT) in neurons; however, increasing evidence has established that mHTT exerts toxic cell effects on other cell types of the CNS, which likely contribute to the pathogenesis of HD as well [6,264,268].

The aggregation of misfolded mHTT is a pathological hallmark of HD and is associated with neuronal loss [6,268]. Astrocytes express mHTT to the same extent as in neurons in HD and its preclinical models [6,268]. The selective overexpression of mHTT in mouse astrocytes is sufficient to recapitulate neurodegeneration and motor symptoms observed in HD and its animal models. Conversely, the selective deletion of reactive astrocytes or astrocyte-specific rescue approaches attenuates neurodegeneration and neurological abnormalities observed in HD models [6,268,269]. In a Drosophila model expressing human mHTT, the downregulation of genes involved in synapse assembly in glial cells mitigates pathogenesis and behavioral deficits [270]. Strikingly, reducing dNRXN3 function in glia is sufficient to improve the phenotype of flies expressing mHTT in neurons [270]. It is now clear that the accumulation or overexpression of mHTT in astrocytes impairs the glutamate uptake and disrupts K+ homeostasis and Ca2+ signaling, leading to astrocyte dysfunction and neurodegeneration [6,268,269]. Brain cholesterol is produced mainly by astrocytes and is important for neuronal function [6,268]. The dysregulation of brain cholesterol homeostasis has been linked to HD [6,268]. The pathogenic impact of cholesterol pathways in HD is supported by a recent study showing that activating the cholesterol biosynthesis pathway in striatal glial cells restores synaptic transmission, clears mHTT aggregates, and attenuates behavioral deficits in an HD mouse model [271]. In addition, a recent study shows that region-specific neuronal toxicity in HD arises from the metabolic reprogramming of astrocytes [272]. Together, astrocyte dysfunction contributes to HD pathogenesis, and targeting astrocyte dysfunction may provide therapeutic potential in HD [6,268].

The role of astrocyte dysfunction in HD pathogenesis is further supported by transcriptomic studies. Transcriptional dysregulation is an early and progressive event that is hypothesized to play an important role in the pathogenesis of HD. Recent snRNA-seq/scRNA-seq studies of astrocytes isolated from post-mortem HD brains and mouse HD models reveal that HD astrocytes experience profound gene expression changes indicative of losing essential normal functions or the activation of inflammatory pathways [273,274]. Further analysis reveals that striatal HD astrocytes display context-specific molecular responses that are regulated by Gi-GPCR activation [275]. Importantly, the selective activation of Gi–GPCR signaling in astrocytes reverses the impairment of synaptic plasticity, Ca2+, and GPCR signaling [275]. In another study, astrocyte-specific transcriptomic analysis shows that the activation of the JAK2-STAT3 pathway in astrocytes coordinates a transcriptional program associated with proteolytic capacity. Importantly, the selective activation of this cascade in astrocytes through viral gene transfer reduces the number and size of mHTT aggregates and improves neurological function in mouse models of HD [276]. These studies identify astrocyte GPCR and JAK2-STAT3 signaling as promising targets to control neurodegeneration in HD.

TGF-β signaling has been implicated in HD [277,278]. Early studies have reported changes in the serum or plasma levels of TGF-β1 in HD patients during disease progression and suggest that TGF-β1 could be used a potential biomarker [279,280,281]. However, a recent study found plasma levels of TGF-β1 in HD patients are not significantly different from the control group and do not change significantly with the progression of the disease [282]. Therefore, the usefulness of plasma TGF-β1 as a biomarker for the assessment of HD severity needs further investigation. In the brain, the expression of TGF-β1 is reduced in cortical neurons in post-mortem brains from HD patients and HD model mice [281], but whether the reduced TGF-β1 expression plays a role in HD pathogenesis is not clear. In the hippocampal stem cell niches of mouse and rat HD models, TGF-β/Smad signaling is elevated, which is considered to contribute to the induction of the quiescence of NSCs leading to reduced hippocampal neurogenesis [278]. The upregulation of the TGF-β pathway is also observed in a human iPSC model of HD, which can be corrected to normal levels by replacing the expanded HTT CAG repeat with a non-pathogenic normal length [283]. Further, the correction of TGF-β signaling pathways reverses disease phenotypes, such as susceptibility to cell death and altered mitochondrial bioenergetics, in iPSC [283]. These results suggest that mHTT activates the TGF-β signaling pathway and that TGF-β signaling plays a disease-promoting role in HD.

A transcriptome analysis of iPSCs derived from HD patients, neural stem cells (NSCs) from HD patients, or striatal cell lines expressing mHTT also reveals TGF-β signaling as the top dysregulated pathway [284,285]. A further network modeling of Smad3 target genes, together with Smad3 expression and phosphorylation, and the Smad3 ChIP-seq of the striatum of HD knock-in mice, identified TGF-β/Smad signaling as a core regulator of early gene expression in HD that has therapeutic implications [286]. Systems-based genetic analyses performed on human transcriptomic datasets from post-mortem HD brains identify a novel astrocyte-specific transcriptional module most relevant to HD pathology [287]. This astrocyte HD module is regulated by TGF-β-FOXO3 signaling [287]. Together, these studies show that TGF-β signaling is an important regulator of HD-associated gene expression. In support of this view, TGF-β signaling has been shown to regulate HTT expression [284] and ameliorate mHTT-induced toxicity [285]. In summary, current evidence suggests that the TGF-β signaling pathway may play an important role in HD pathogenesis. However, it is important to note that no studies have directly tested whether manipulating TGF-β signaling alters the disease outcome in conventional animal models of HD. Therefore, further studies are needed to determine if TGF-β signaling is a potential therapeutic target for HD.

### 5.9. Epilepsy

Epilepsy is a common neurological disorder characterized by an enduring predisposition to generate epileptic seizures [288,289]. Epilepsy is a major public health problem, affecting approximately 1% of the total population worldwide [288]. Temporal lobe epilepsy (TLE) is the most prevalent form of epilepsy and many patients with TLE develop drug resistance [290]. Epilepsy may occur as a result of brain injury, stroke, tumors, infections (meningitis or encephalitis), autoimmune diseases, or genetic mutations [288,289]. The underlying mechanism of epileptic seizures is excessive and abnormal neuronal activity in the cortex of the brain [291]. Recent research points towards the role of non-neuronal cells such as astrocytes in the genesis and spreading of seizures in the brain [289,292,293].

Reactive astrocytes are found both in animal models of epilepsy and in brain tissue from patients with seizures [288]. Astrocytes express ion channels, transmitter receptors, and transporters, and can thus sense and respond to neuronal activity [294]. Many molecular alterations during astrogliosis are causally linked to epileptogenesis, including the downregulation of gap junction connexins, glutamate transporters, potassium channels and aquaporin 4 channels, as well as the activation of inflammatory pathways [288]. Recent studies have shown that the C3-C3aR pathway is involved in epilepsy and epilepsy-associated neurodegeneration [295]. mRNA and protein levels of C1q and C3 are increased in the hippocampus of patients with TLE [296]. Similar results are also obtained from animal models. C3-positive astrocytes are significantly increased in the diisopropylfluorophosphate rat model of epilepsy [297]. C1q-C3 signaling activation in status epilepticus correlates with epileptic seizure frequency [296], and blocking C3 signaling reduces seizure activity and neuronal injury [298,299]. Since C3 is principally produced by activated astrocytes [45], these results further support the epileptogenic role of reactive astrocytes in epilepsy. In agreement with this view, it was recently suggested that astrocyte reactivity may be used as a biomarker for epilepsy [289].

TGF-β signaling is activated and components of the pathway are upregulated in human and experimental epilepsy. TGF-β1 levels are increased in the CSF of patients with drug-resistant epilepsy [300]. TGFBR1 expression is significantly upregulated in temporal neocortices of patients with TLE [301]. Single nucleotide polymorphisms in TGFBR1 are associated with a risk of epilepsy in a Chinese population [302]. More specifically, the TGFBR1 AT and TT genotypes emerge as a protective factor, whereas the TCTAT and TCCAA haplotypes emerge as a risk factor for epilepsy [302]. Further qRT-PCR analysis shows that TGFBR1 mRNA levels are significantly higher in epilepsy patients than in controls. Genotype–phenotype analysis show lower levels of TGFBR1 mRNA in carriers with the rs6478974 TT genotype, which is confirmed by eQTL data [302]. Biallelic loss-of-function mutations in TGFB1 result in very early-onset inflammatory bowel disease and CNS dysfunction associated with epilepsy, brain atrophy, and posterior leukoencephalopathy [303]. The expression of the Smad anchor for receptor activation (SARA) and the level of p-Smad3 are upregulated in the brain of an epileptic rat model, as well as in the temporal cortex of patients with TLE [304]. We have shown that the systemic administration of kainic acid, an epileptogenic and neuroexcitotoxic agent [305], results in a rapid and persistent activation of TGF-β signaling in the SBE-luc reporter mice [21]. Recently, microRNAs (miRNAs) have emerged as promising therapeutic targets for seizure control and disease modification. Veno et al. performed small RNA-seq of Ago2-loaded miRNAs from three different seizure models and identified dysregulated and functionally active miRNAs in seizure pathogenesis [306]. Combinatorial miRNA inhibition (by combining the three most effective antagomirs targeting miR-10a-5p, miR-21a-5p, and miR-142a-5p) reduced seizures in experimental TLE. Interestingly, target and pathway analysis revealed a role of the TGF-β signaling pathway in the anti-seizure effects of combinatorial miRNA inhibition [306]. This evidence suggests that TGF-β signaling is involved in the generation of seizure activity. In line with these observations, TGF-β pathway proteins are identified as key regulators driving epileptogenesis in TLE by a recent study of systems-level analysis [307].

TGF-β has been shown to increase neuronal excitability and trigger epileptogenesis and seizures [308]. Incubating cortical slices with TGF-β1 directly induces epileptiform activity and epileptogenic transcriptional responses, as well as astrocyte reactivity and inflammation [69]. This effect may be related to TGF-β’s upregulation of IL-6, which causes neuronal hyperexcitability and epileptiform discharges in vitro and spontaneous seizures in mice [68]. TBI is a common cause of acquired epilepsy [308,309,310]. The role of TGF-β signaling in post-traumatic seizures is supported by the observation that the inhibition of TGF-β signaling with a Tgfbr1 inhibitor, LY-364947, significantly reduces the duration and severity of post-traumatic seizures [311]. Mechanistically, post-traumatic seizures are mediated via astrocytic TGF-β signaling activated by albumin [312,313]. Animal studies have demonstrated that albumin extravasation in epilepsy is due to BBB breakdown [312,314,315]. When the BBB function is compromised, albumin enters the brain’s extracellular space, accumulates in perivascular astrocytes, and binds to TGFBR2, leading to the activation of astrocytic TGF-β signaling [314]. The activation of TGF-β signaling then causes astrocyte reactivity, with an impairment of potassium buffering and glutamate metabolism, and disruption of water homeostasis. All of these changes promote neuronal hyperexcitability and spontaneous seizures [289,312,313]. The role of albumin in epilepsy has been demonstrated in infusion studies. The intracerebroventricular infusion of albumin activates astrocytic TGF-β signaling and induces astrocyte reactivity and excitatory synaptogenesis that precedes the development of spontaneous seizures [64,316]. Similar results are also obtained for ex vivo studies. The arterial perfusion of albumin exacerbates bicuculline methiodide-induced epileptiform seizure-like events and astrogliosis in isolated guinea pig brain [317]. Strikingly, BBB breakdown, albumin, and TGF-β1 exposure provoke a similar hypersynchronous neuronal epileptiform activity and epileptogenic transcriptional response, including the activation of NF-κB, Jak-Stat, MAPKKK, and the complement signaling pathways [69]. Notably, blocking TGF-β signaling with Alk5 inhibitors SJN2511 or SB431542, or with angiotensin receptor 2 antagonist losartan, prevents the albumin-initiated gene response and epilepsy [69,314,316,318,319]. Therefore, astrocyte TGF-β signaling triggers hyperexcitability and seizures, and targeting TGF-β signaling could be a feasible strategy for the disease modification and prevention of epilepsy [315,320]. In addition to epilepsy, BBB breakdown contributes to neuronal hyperexcitability in aging and AD [66,319,321]. In patients with epilepsy or AD, BBB impairments are spatially associated with an electroencephalogram (EEG) signature of a transient slowing of the cortical network, termed paroxysmal slow wave events [321]. The infusion of albumin directly into the cerebral ventricles of naïve young rat results in a high incidence of this transient EEG abnormality, which is also observed in aged mice and models of AD and epilepsy [321].

In support of the critical role of TGF-β signaling in epilepsy, several studies have discovered the downstream target gene CDKN1A as a central hub in epileptogenesis. A transcriptional analysis of rat piriform cortex following sarin-induced seizures identified several significant canonical pathways and de novo networks of genes most significantly modulated by seizure. Two of the five most significant networks identified are built around TGF-β and Cdkn1a [322]. An integrative analysis of epilepsy animal models and human epilepsy tissue found five key genes, including TGF-β and CDKN1A, as central nodes in the protein networks in epileptogenesis [323]. mRNA levels of TGF-β and CDKN1A are upregulated in the cortex of patients with epilepsy [323]. Exosomes from epileptogenic tissue cause the induction of key pathways in cultured cells, including the inflammatory response and key signaling nodes SQSTM1 (p62) and CDKN1A (p21) [324]. Recently, dysregulated miRNA expression has been associated with epileptogenesis through inflammatory pathways, cell death, neuronal excitability, and synaptic reorganization [325]. TGF-β and CDKN1A-related pathways are common targets of epilepsy-associated miRNA [325]. These studies support a central role of CDKN1A in epileptogenesis. However, there has been no report on the exact role of CDKN1A in epilepsy, and CDKN1A has not been investigated in the context of TGF-β signaling.

## 6. Opportunities and Challenges of Targeting TGF-β Signaling in the Brain

As discussed above, the manipulation of the TGF-β signaling pathway in astrocytes alters the disease outcome. Findings from these studies not only help our understanding of disease pathogenesis but also have therapeutic implications. Depending on the role of TGF-β signaling in the disease (disease-promoting vs. mitigating), strategies of both the activation and inhibition of TGF-β signaling have been explored in brain injury and disease.

To harness the therapeutic potential of neuroprotective and anti-inflammatory effects, TGF-β must be delivered into the brain. However, TGF-β does not cross the intact BBB; thus, it requires intracerebral administration for TGF-β to be effective. The ICV delivery of TGF-β1 has been tested in animal models of TBI [114], stroke [131], AD [326], and PD [207]. The ICV route of drug delivery in general is a valuable option for achieving a high local drug concentration in a target brain region while minimizing systemic toxicity [327]. However, it requires an invasive procedure, with difficulties in providing long-term administration and risks of complications, such as infection [327], thereby limiting its clinical application. To overcome these shortcomings, several non-invasive strategies have been developed, including focused ultrasound [328] and nanoparticles [329]. It should be noted that it is almost impossible to achieve cell-type-specific effects when using the ICV delivery of TGF-β ligands, because the receptors TGFBR1 and TGFBR2 are widely expressed among many cell types in the brain. Adeno-associated virus (AAV) vectors have been shown to be safe and effective in targeting glial cells and have been used in clinical trials [330]. We have used AAV encoding a constitutively active form of Tgfbr1 to activate TGF-β signaling [209]. A number of studies have used different AAV serotypes with astrocyte-specific promoters to increase gene expression in astrocytes [163,276]. Excitingly, the newly developed AAV-PHP.B provides a non-invasive alternative for gene delivery to the brain through systemic injection [331]. AAV-PHP.B is able to transduce the majority of astrocytes in multiple brain regions. However, these approaches have not been used to manipulate TGF-β signaling, and true astrocyte-cell specific targeting remains elusive [330].

Based on the findings that the deficiency of TGF-β signaling in neurons promotes neurodegeneration in the context of AD and PD [179,209], we hypothesize that neuronal TGF-β signaling plays a protective role in AD and PD and that the rescue of neuronal TGF-β signaling has therapeutic potential. This hypothesis is supported by the findings that activating TGF-β signaling reduces MPTP-induced dopaminergic neurodegeneration and motor deficits [209]. These findings motivate us to screen and develop small molecule TGF-β/Smad activators. We have developed a novel small molecule drug, C381. C381 is orally bioavailable and brain-penetrant, and has exhibited great potential for clinical translation [26]. Meanwhile, other groups have proposed using drugs that cross the blood–brain barrier and are potentially able to activate TGF-β signaling [332]. Nutraceuticals, such as Hypericum perforatum (hypericin and hyperforin), flavonoids such as hesperidin, omega-3, and carnosine, can increase TGF-β1 production in the brain [332]. Some of these may have synergism with currently approved cognition-enhancing medication, and thus represent a novel pharmacological approach in improving cognitive function [332].

In disease conditions where TGF-β signaling is activated and the outcome of signaling is diverted toward disease progression, blocking TGF-β signaling provides therapeutic benefits. The development of therapeutic approaches of blocking TGF-β signaling has mainly been driven by motivations to inhibit the progression of cancer and fibrosis [333,334,335]. Different strategies have been developed to block TGF-β signaling (Figure 1), including antisense oligonucleotides that abrogate the expression of the TGF-β ligand and its receptor, small molecules or antibodies that selectively interfere with TGF-β activation, monoclonal neutralizing antibodies against the TGF-β ligand and its receptor, ligand traps that sequester TGF-β and prevent its receptor binding, and small-molecule inhibitors of the TGF-β receptor kinases [333,334,335]. TGFBR1 (ALK5) is an attractive target due to its druggability, as well as specificity, in the pathway. A number of potent, selective inhibitors have been developed [333,334,335]. Many of them have advanced to clinical trials and demonstrated acceptable safety profiles and therapeutic effects in cancer treatment [336]. Some have been tested and shown beneficial effects in preclinical models of aging [66], ALS [238], and EAE [22]. Pirfenidone, an FDA-approved drug for treating fibrosis, inhibits TGF-β1 production and its downstream pathways [337] and has been shown to reduce neurodegeneration and neuroinflammation after kainic acid injury [338] and TBI [339]. These small molecule drugs have advantages, in that most can be administered orally, and some can pass BBB. However, none of the anti-TGF-β drugs have been tested in clinical trials for brain injury and neurodegeneration.

Losartan has gained great interest in targeting TGF-β’s detrimental effects because it is an FDA-approved anti-hypertension drug. Losartan has been shown to ameliorate disease in TBI [118] and EAE [22,63]. For epilepsy, in addition to the epilepsy models involved in BBB breakdown and albumin extravasation [64,318,319], losartan has also been shown to attenuate seizure activity and neuronal damage in other models of epilepsy by different research groups [340,341,342]. A recent publication suggests that telmisartan, another angiotensin receptor 2 antagonist with anti-TGF-β effects, has the potential to reduce seizure frequency when administered as an add-on antiepileptic drug in dogs with refractory idiopathic epilepsy [343]. However, losartan fails to suppress seizure-like activity in cortical and hippocampal areas of human brain slices of patients with drug-resistant TLE, suggesting that further exploration may be required for losartan as an anticonvulsant drug in clinical trials [344]. The biggest hurdle for losartan clinical translation is the knowledge gap in dosing recommendations. It is not known whether dosing humans with the equivalent doses of losartan used in preclinical studies is necessary and feasible [345]. The dose range used in preclinical studies (1–100 mg/kg/day) is equivalent to 68–978 mg/day for a 60 kg person, whereas the FDA-allowed max dose is 100 mg/day [345]. Further dose–response studies are needed to determine whether human studies are feasible.

In summary, a number of strategies are now available to manipulate the TGF-β signaling pathway in the brain, and they have been demonstrated to able to improve the disease outcome in preclinical studies, but additional research is needed in order for them to be tested in clinical trials.

## 7. Conclusions and Future Directions

Astrocytes regulate multiple essential processes in the nervous system in normal and disease conditions. Astrocyte reactivity is not only a universal response of astrocytes to brain injury, aging, and age-related neurodegenerative diseases, but also a central mechanism driving the development and/or progression of these conditions. The response that astrocytes elicit is highly context- and disease-dependent. Reactive astrocytes are highly heterogeneous and constitute subpopulations with unique molecular signatures depending on the brain insult, disease state, and distance from primary lesions [1,2,3,4,5,6,7,8]. Furthermore, astrocyte reactivity is a dynamic process that could be normalized or even reversed. The modulation of the astrocyte reactivity state has emerged as an important venue for new preventive and therapeutic strategies. However, attempts to block astrocyte reactivity globally have yielded inconsistent effects on functional outcomes. This is likely, at least in part, due to the heterogeneity of the astrocyte response to injury and disease. Different astrocyte subpopulations/subsets seemingly coexist in reactive astrogliosis; however, the source and regulation of such heterogeneity are not completely understood [346]. Therefore, it is of high importance to identify and target the astrocyte subsets that acquire maladaptive functions and are “harmful” for the diseased brain. Future research will aim to identify molecular pathways that drive beneficial and detrimental phenotypes, which will ultimately facilitate the development of pathway-specific therapeutic approaches to promote the beneficial effects while downregulating the harmful/maladaptive effects of reactive astrocytes [33]. The integration of advanced sequencing technologies (single-cell and spatial transcriptomics) with multi-omics approaches will facilitate the discovery of novel mechanisms for therapeutic intervention.

TGF-β signal transduction is overwhelmingly complex and diverse due to the large numbers of interacting components, as well as the complicated feedback and crosstalk with other pathways. The intensity and duration of TGF-β signaling are tightly regulated both spatially and temporally, at multiple levels (Figure 1), reflecting the remarkable delicacy, specificity, and context dependency of TGF-β functions. In the brain, TGF-β plays fundamental roles in development and homeostasis in normal conditions, as well as in inflammation and repair following injury and neurodegeneration. For astrocytes, TGF-β is a powerful regulator and effector of astrocyte reactivity. TGF-β regulates many aspects of astrocyte function and induces major changes in response to injury and aging. The genetic and pharmacological manipulation of the TGF-β signaling pathway in astrocytes alters disease outcomes in many preclinical models of CNS injury and disease, which present unique opportunities for the discovery and development of novel therapeutics. Despite an abundance of literature and significant progress to date, there are still outstanding questions remaining in the field. First, the precise mechanisms of TGF-β signaling in brain injury and neurological diseases remain to be elucidated. In particular, the downstream effectors and target genes have not been identified in most disease conditions. A thorough understanding of TGF-β’s role at different disease stages is an urgent need for the development of effective, precise treatment regimens targeting TGF-β. Second, given the context- and cell-type-dependent nature of TGF-β function, it is critical to manipulate this signaling pathway in a cell-type-specific manner. However, it remains a challenge to achieve cell-type-specific manipulation with current TGF-β targeting strategies. This review wishes to provide a basis for future research aimed at gaining more mechanistic insights into astrocyte reactivity and TGF-β signaling, which are essential for the understanding and fine-tuning of the TGF-β signaling pathway in order to develop targeted therapies.

## Figures and Tables

**Figure 1 biomedicines-10-01206-f001:**
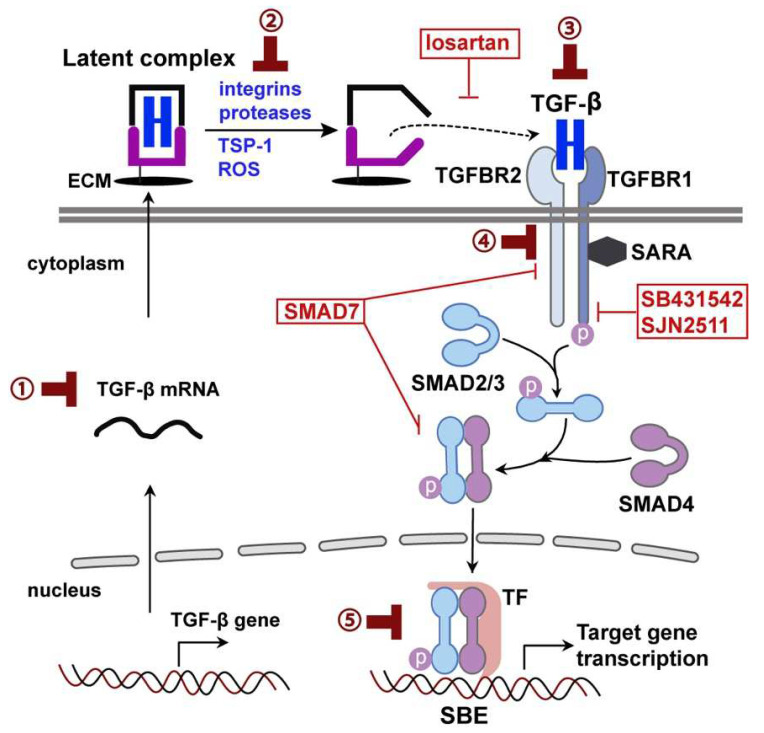
TGF-β activation and the canonical, SMAD-dependent signaling pathway. TGF-β is produced in a latent form as part of a latent complex tethered to ECM. The latent complex consists of mature dimeric TGF-β, associated with latency-associated peptide (black color) and a latent TGF-β-binding protein (magenta color). Latent TGF-β can be activated and released from the complex by integrins, proteases, TSP-1, and ROS. Active TGF-β binds to its receptors and initiates Smad signaling to exert its biological effects. The activities of TGF-β signaling can be modulated through (1) TGF-β translation and production, (2) TGF-β activation, (3) TGF-β neutralization using recombinant antibodies, (4) synthetic molecules that inhibit the phosphorylation of the TGFBR and SMAD, and (5) targeting downstream effectors. The red boxes show inhibitors and their action sites mentioned in this review.

**Figure 2 biomedicines-10-01206-f002:**
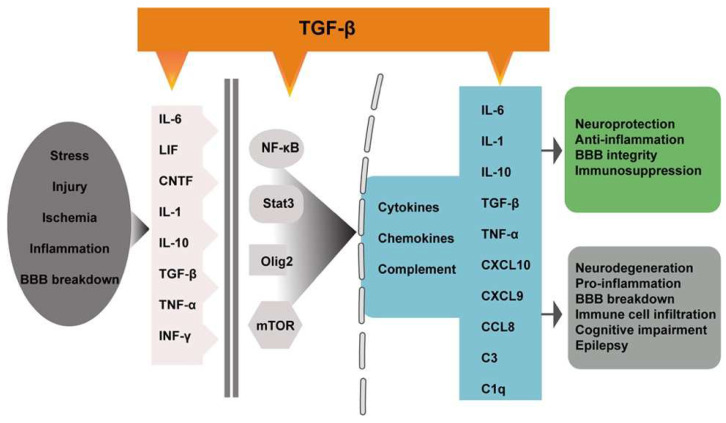
Regulators and signaling pathways of astrocyte reactivity. In response to insults such as stress, injury, ischemia, inflammation, and BBB breakdown, brain cells produce inflammatory factors that trigger astrocyte reactivity through transcriptional pathways involving NF-κB, Stat3, Olig2, and mTOR. These pathways regulate the production of cytokines, chemokines, and complements, which mediate the neuroprotective or neurodegenerative effects of reactive astrocytes, and are also involved in triggering and maintaining astrocyte reactivity. TGF-β regulates astrocyte reactivity through multiple mechanism: inflammatory factors, transcriptional pathways, and downstream target genes.

## Data Availability

Not applicable.

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
