# Peer review of "TGF-β as a Key Modulator of Astrocyte Reactivity: Disease Relevance and Therapeutic Implications"

_biomedicines, 2022, doi:10.3390/biomedicines10051206_

Round 1
Reviewer 1 Report
The work presents an overview of the role of TGF-beta in various branches of neurology. Authors refer to more common clinical entities, however there are some points which should be reorganized:
- Authors refer to most most common neurodegenerative entities e.g. AD PD, ALS, however in my opinion regarding the ongoing discussion on the pathogenesis of Parkinsonisms and their association, it would be worth mentioning that the inflammatory mechanisms in PD and atypical parkinsonisms tend to differ. A separate paragraph regarding atypical Parkinsonisms should be added: Ref >>>>> TGF-beta receptors-I and -II immunoexpression in Alzheimer's disease: a comparison with aging and progressive supranuclear palsy. Neurobiol Aging. 1998 Nov-Dec;19(6):527-33. doi: 10.1016/s0197-4580(98)00089-x. PMID: 10192211. >>>>> Microglial Activation and Inflammation as a Factor in the Pathogenesis of Progressive Supranuclear Palsy (PSP). Front Neurosci. 2020;14:893. Published 2020 Sep 2. doi:10.3389/fnins.2020.00893 >>>>> Platelet-to-lymphocyte ratio and neutrophil-tolymphocyte ratio may reflect differences in PD and MSA-P neuroinflammation patterns. Neurol Neurochir Pol. 2022;56(2):148-155. doi: 10.5603/PJNNS.a2022.0014. Epub 2022 Feb 4. PMID: 351186335118638
- A wider elaboration on clinical implications should be added.
- Authors should also give their point of view in the context of the publications mentioned in the review as currently the work may seem more like listing the works. A view on future perspectives should be also added.
Author Response
- 1. Authors refer to most most common neurodegenerative entities e.g. AD PD, ALS, however in my opinion regarding the ongoing discussion on the pathogenesis of Parkinsonisms and their association, it would be worth mentioning that the inflammatory mechanisms in PD and atypical parkinsonisms tend to differ. A separate paragraph regarding atypical Parkinsonisms should be added: Ref >>>>> TGF-beta receptors-I and -II immunoexpression in Alzheimer's disease: a comparison with aging and progressive supranuclear palsy. Neurobiol Aging. 1998 Nov-Dec;19(6):527-33. doi: 10.1016/s0197-4580(98)00089-x. PMID: 10192211. >>>>> Microglial Activation and Inflammation as a Factor in the Pathogenesis of Progressive Supranuclear Palsy (PSP). Front Neurosci. 2020;14:893. Published 2020 Sep 2. doi:10.3389/fnins.2020.00893 >>>>> Platelet-to-lymphocyte ratio and neutrophil-tolymphocyte ratio may reflect differences in PD and MSA-P neuroinflammation patterns. Neurol Neurochir Pol. 2022;56(2):148-155. doi: 10.5603/PJNNS.a2022.0014. Epub 2022 Feb 4. PMID: 351186335118638
Thank you for your suggestion and for providing the references. A new paragraph on atypical Parkinsonisms is added and all references are cited.
- 2. A wider elaboration on clinical implications should be added.
I added a new section discussing clinical implications ("6. Opportunities and challenges of targeting TGF-β signaling in the brain"). Other changes include: adding "Disease Relevance and Therapeutic Implications" to the title to make it more specific and adding details on clinical implications when discussing TGF signaling in each disease condition.
- 3. Authors should also give their point of view in the context of the publications mentioned in the review as currently the work may seem more like listing the works. A view on future perspectives should be also added.
I agree this is one weakness of this review. I revised the whole review and added a more derails in Section 7 "Conclusions and future directions".
All changes are highlighted in blue in the revised manuscript.
Thank you for the comments.
Reviewer 2 Report
I just have minor suggestions. 1) Please provide summary sentence in 5.4. Alzheimer's disease section. 2) Please consider adding section regarding "Huntington's disease" which is a neurodegenerative disease and whose pathological progression is closely associated with TGF-β and reactive astrocyte.
Author Response
- 1) Please provide summary sentence in 5.4. Alzheimer's disease section.
I revised and provided a summary sentence at the end of 5.4.
- 2) Please consider adding section regarding "Huntington's disease" which is a neurodegenerative disease and whose pathological progression is closely associated with TGF-β and reactive astrocyte.
I added a new section on "Huntington's disease".
Thank you for reviewing and the suggestions. All changes are marked in blue in the revised manuscript.
Round 2
Reviewer 1 Report
Authors have implemented most of the suggested changes, however I believe that the paragraph regarding atypical Parkinsonisms may be misleading:
"
The pathogenesis of Atypical Parkinson disorders is not well understood, but current evidence suggests the inflammatory mechanisms in PD and Atypical parkinson- isms appear to differ [215]. Pathologically, PSP and CBD are tauopathies. PSP is charac- terized by tau-enriched tufted astrocytes and NFTs in subcortical nuclei [214,216]. The pathologic features for CBD are cortical and striatal tau-positive neuronal and glial le- sions, especially astrocytic plaques and thread-like lesions, along with neuronal loss in focal cortical regions and in the substantia nigra [214,216]."
I think authors should emphasize that the boundaries between PSP and CBD are often questionable. Both pathologies show overlapping clinical manifestations e.g. syndromes PSP-CBS, CBD-PSP etc. This should be stressed in the context of the discussion concerning their pathophysiology.
Ref.
- Neutrophil-to-lymphocyte ratio (NLR) at boundaries of Progressive Supranuclear Palsy Syndrome (PSPS) and Corticobasal Syndrome (CBS). Neurol Neurochir Pol. 2021;55(1):97-101. doi: 10.5603/PJNNS.a2020.0097. Epub 2020 Dec 14. PMID: 33315235.
- Differential Diagnosis of Rare Subtypes of Progressive Supranuclear Palsy and PSP-Like Syndromes-Infrequent Manifestations of the Most Common Form of Atypical Parkinsonism. Frontiers in aging neuroscience, 14, 804385. https://doi.org/10.3389/fnagi.2022.804385
- Genetic pleiotropy and the shared pathological features of corticobasal degeneration and progressive supranuclear palsy: a case report and a review of the literature. Neurocase. 2021 Apr;27(2):120-128. doi: 10.1080/13554794.2021.1879869. Epub 2021 Mar 23. PMID: 33754963; PMCID: PMC8137543.
Author Response
- I think authors should emphasize that the boundaries between PSP and CBD are often questionable. Both pathologies show overlapping clinical manifestations e.g. syndromes PSP-CBS, CBD-PSP etc. This should be stressed in the context of the discussion concerning their pathophysiology.
This point is well taken. I added discussion as suggested and cited suggested references. Thank you for pointing this out.
Jian
Round 3
Reviewer 1 Report
I do not have further comments.